# Textile One-Component Organic Electrochemical Sensor for Near-Body Applications

**DOI:** 10.3390/mi13111980

**Published:** 2022-11-15

**Authors:** Rike Brendgen, Carsten Graßmann, Sandra Gellner, Anne Schwarz-Pfeiffer

**Affiliations:** 1Research Institute for Textile and Clothing (FTB), Niederrhein University of Applied Sciences, Webschulstr. 31, 41065 Moenchengladbach, Germany; 2Faculty Electrical Engineering and Computer Science, Niederrhein University of Applied Sciences, Reinarzstr. 49, 47805 Krefeld, Germany; 3Faculty of Textile and Clothing Technology, Niederrhein University of Applied Sciences, Webschulstr. 31, 41065 Moenchengladbach, Germany

**Keywords:** textile sensors, biosensing, smart textiles, porous electrode, flexible transistor, OECT

## Abstract

The need for more efficient health services and the trend of a healthy lifestyle pushes the development of smart textiles. Since textiles have always been an object of everyday life, smart textiles promise an extensive user acceptance. Thereby, the manufacture of electrical components based on textile materials is of great interest for applications as biosensors. Organic electrochemical transistors (OECTs) are often used as biosensors for the detection of saline content, adrenaline, glucose, etc., in diverse body fluids. Textile-based OECTs are mostly prepared by combining a liquid electrolyte solution with two separate electro-active yarns that must be precisely arranged in a textile structure. Herein, on the other hand, a biosensor based on a textile single-component organic electrochemical transistor with a hardened electrolyte was developed by common textile technologies such as impregnation and laminating. Its working principle was demonstrated by showing that the herein-produced transistor functions similarly to a switch or an amplifier and that it is able to detect ionic analytes of a saline solution. These findings support the idea of using this new device layout of textile-based OECTs as biosensors in near-body applications, though future work must be carried out to ensure reproducibility and selectivity, and to achieve an increased level of textile integration.

## 1. Introduction

“Smart textiles” is a term used to describe textiles that exceed their usual functionalities. As per the definition by the International Organisation for Standardisation (ISO), smart or intelligent textiles are systems consisting of textile and non-textiles components that actively interact with their environment, a user, or an object [1]. Smart textiles have become important as they offer possible applications in numerous industry sectors, such as sports [2], health [3,4], home and living [5], and many more. The growth is especially driven by textile-based sensors used to monitor vital signals for medical and sports purposes. The need for more efficient health services and increasing awareness about a healthy lifestyle contribute to this development. Thereby, textile-based electronics promise extensive user acceptance since textiles have always been an object of everyday life. Furthermore, the growing usage of wireless technology and the miniaturization of electronics contribute to this trend. One challenge hereby is the integration of sensors into the textile material to ensure comfort and wearability [6,7].

Textile-based organic electrochemical transistors (OECTs) can help in this regard. They consist of a gate electrode, an electrolyte, and a source/drain electrode, and often find application in biosensors [8,9,10,11,12]. One big advantage of OECTs is their operation at very low voltages, which is suitable for near-body biosensing applications [13]. By applying a voltage to the gate electrode, the current flow between the source and the drain electrode can be controlled. Thereby, the presence of an electrolyte enables the flow of ions into the channel of the source/drain electrode, and thus changes the current flow [13,14,15]. For this mechanism, the source/drain electrode needs to be made of an electrically conductive polymer as the basic working principle relies on the doping and dedoping of the conductive polymer, which results in differences in channel conductivity. The changes in the doping state originate from the injection of ions from the electrolyte into the polymer [16]. The electrically conductive polymer poly(3,4-ethylenedioxythiophene) polystyrene sulfonate (PEDOT:PSS) is of interest for applications as electrodes in electrical and chemical transistors, as well as for electrocardiographs, for organic solar cells or organic light-emitting diodes. Next to its high electrical conductivity, PEDOT:PSS also possesses high transparency in visible light, high mechanical flexibility, high thermal stability and good oxidation resistance [6,17,18,19,20]. Ultimately, the changes in ion intensity can be detected in the current flow between the source and drain (channel) electrode when the electrolyte is enriched with some sort of ions coming from diverse body fluids. This conduct favors the use as a biosensor [13,14,16]. OECTs based on PEDOT:PSS work in depletion mode, which means that in absence of a positive gate voltage, a hole current flows in the channel (ON-state). When applying a positive gate voltage, ions in form of cations (positively charged) are injected from the electrolyte into the channel and the negatively charged ions (anions) of the PEDOT:PSS become saturated. Consequently, the number of mobile holes in the channel decreases, the conductive polymer is dedoped, the drain current drops and the transistor reaches its off state [16]. Several studies describe textile OECT arrangements, where the gate electrode and source/drain electrode are electro-active yarns that are distinctly separated from each other and only connected by a liquid, gel-like electrolyte solution [21,22,23,24,25]. I Gualandi et al. (2016) presented a printed textile-based OECT, but here again, the electrolyte was externally applied [26], which limited the scope of application to moist environments. This paper, on the other hand, presents an OECT where the three components—gate electrode, electrolyte, and source/drain electrode—are combined in one device so that integration into a garment for future applications as biosensors becomes easier, mostly because the electrolyte is not externally applied. In this way, the herein-developed OECT can easily be integrated into a textile and function as a switch or an amplifier, without additionally applied electrolytes. Therefore, a solid electrolyte layer that distinctly separates, but at the same time permanently connects, the gate and source/drain electrode and still allows the injection of ions into the channel of the source/drain electrode needs to be established. Moreover, a porous layer of PEDOT:PSS was developed that allows diffusion of ionic analytes through the porous source/drain electrode and enriches the electrolyte with ions when an analyte comes in contact with the transistor. At a constant gate and source/drain voltage, this change in ion intensity becomes noticeable by a decrease in current output measured at the source/drain electrode.

## 2. Materials and Methods

The following sections describe the materials used and the procedure to manufacture and test this textile-based one-device OECT. Firstly, the materials used will be stated, while secondly the manufacturing steps will be described and lastly characterization methods are highlighted.

### 2.1. Materials

For the different components (gate electrode, source/drain electrode, and electrolyte) of an OECT, various materials were used that will be described.

#### 2.1.1. Gate Electrode

The gate electrode was simply built from a nickel/copper-coated ripstop fabric, provided by LessEMF, USA. According to the datasheet, it had a thickness of 0.08 mm and an aerial weight of 90 g/m^2^. The electrical surface resistance is given at 0.03 Ohm/m^2^. It was lightweight and flexible, easy to cut and sew, similar to ordinary fabric [27].

#### 2.1.2. Electrolyte

The electrolyte layer was made of an electrode contact gel and hydroxyethyl cellulose (tylose) mixed with glycerol and lithium chloride (LiCl) as an ionic salt. The electrode contact gel was chosen as the base material as it ensures conductivity during ECG and EG examination and contains ions that function as charge carriers. A commercially available electrode contact gel was purchased from P.J. Dahlhausen & Co. GmbH, Germany. Tylose type H 60000 YP2 (SE Tylose GmbH & Co. KG, Wiesbaden, Germany) was used as the binding agent for the electrolyte layer while functioning as a liquid retention agent at the same time. Moreover, glycerol 99.5% (VWR International GmbH, Darmstatd, Germany) was added because of its water-binding and moisturizing properties. Lithium chloride was used as the ionic salt that builds strong hygroscopic crystals and was supplied by Alfa Aesar (Thermo Fisher (Kandel) GmbH, Kande, Germany).

#### 2.1.3. Source/Drain Electrode

The porosity of the source/drain electrode was achieved by impregnating a cellulose-based, biodegradable micro filtering paper (nonwoven) for coffee-making with PEDOT:PSS. Because of its commercial application, it was considered to be porous, which has been proven by the microscopic examination in Section 3.1. The nonwoven textile had a diameter of 6.3 cm but it was cut into strips of 1 cm width so that it could be vertically immersed in the PEDOT:PSS solution. The PEDOT:PSS solution consisted of a 1.3 wt% PEDOT:PSS dispersion (Sigma-Alderich Chemie GmbH, Taufkirchen, Germany) mixed with dimethyl sulfoxide (DMSO)(Sigma-Alderich Chemie GmbH, Taufkirchen, Germany) as the secondary dopant and glycerol 99.5% (VWR International, Darmstadt, Germany) as a water-binding agent.

### 2.2. Methods

The OECT was manufactured in several steps, whereby only the nickel–copper fabric stayed untreated. The above-mentioned materials for the different components of the OECT (electrolyte and source/drain electrode) were separately mixed before bringing the components together in the one-device OECT.

#### 2.2.1. Electrolyte

The first step for the preparation of the electrolyte was mixing 5% tylose with distilled water and stirring it for 2 h in the dispermat VMA Getzmann GmbH, Germany. Secondly, the diluted tylose was mixed with the commercial ECG-gel in a ratio of 1:1 by hand. Subsequently, 10 wt% glycerol and 1 wt% LiCl were added to this base mixture, and all was stirred for 20 min in the dispermat. As soon as the solution started to form bubbles, one drop of defoamer Lyoprint Air (Huntsman Textile Effects GmbH, Langweid am Lech, Germany) was added to remove the air.

#### 2.2.2. Source/Drain Electrode

For the production of the porous semi-conducting source/drain electrode, firstly the PEDOT:PSS solution needed to be prepared. Therefore, PEDOT:PSS, glycerol and DMSO were mixed in a ratio of: 17.25, 10.12, and 72.63 wt%, respectively. This composition was based on the experimental set-up of Malti et al. [14]. Therein, a nanofibrillated cellulose, glycerol, and DMSO-blended PEDOT:PSS film is presented, and the above-described ratio was derived from the original composition of 16.2 wt% PEDOT:PSS, 6.1 wt% nano cellulose, 68.2 wt% DMSO, and 9.5 wt% glycerol [14]. Instead of using diluted nanocellulose, in the underlying work, a porous nonwoven fabric was applied so that the corresponding weight percentage of nanocellulose was neglected, which resulted in the above-mentioned weight ratio of 17.25, 10.12, and 72.63 wt% (PEDOT:PSS, glycerol, and DMSO). The solution was stirred by a magnetic stirring rod and subsequently placed into an ultrasound bath for 30 min. Next, the filtering paper was vertically immersed into the PEDOT:PSS mixture and left to absorption for at least 4.5 h, then the immersed paper was dried at 50 °C for approximately 2.5 h.

#### 2.2.3. Assembly of the OECT

All three components—the nickel–copper woven textile, PEDOT:PSS-immersed nonwoven fabric, and electrolyte solution—were brought together by laminating with squeegees using the electrolyte as the adhesive material. A hand squeegee with a wet-film thickness of 500 μm was used to apply the electrolyte solution to the nickel–copper fabric, while afterwards the PEDOT:PSS-coated nonwoven fabric was placed in the still liquid electrolyte solution. The laminate was dried in the oven at 60 °C for 10 min so that the electrolyte solution hardened and thereby bonded all three layers. After drying, the single OECTs were cut out with a pair of scissors, whereby the whole dimensions comprised 3 cm in length and 1 cm in width (Figure 1).

#### 2.2.4. Characterization

After the preparation of the one-component OECTs, optical inspections and electrical characterization were performed. The former comprises the analysis of the whole OECT structure as well as of only the uppermost layer—the source/drain electrode—with a scanning electron microscope (SEM) (TM4000Plus, Hitachi High-Tech Corporation, Tokyo, Japan). The source-drain electrode was examined for its porosity and successful coating and was compared with the untreated nonwoven textile. The composed transistor was investigated for its layered structure. In addition, energy dispersive X-ray spectroscopy (EDX) (Bruker Corporation, Billerica, MA, USA) was carried out to identify the chemical elements of the individual components. The porosity of the coated and uncoated nonwoven material was investigated with a PSM 165 capillary fluid pore size meter from Topas (Dresden, Germany). The samples were measured dry and wetted with Topor, which is a special test fluid for this method and mainly contains perfluoro tri ***n***-butylamine and isobutyl perfluoro ***n***-butylamine. In principle, the test fluid fills all pores of the specimen. When the gas flows through, the specimen increases and the pores become gas permeable at a certain point, which is the bubble point of the material and corresponds to the opening pressure of the largest pore. By further increasing the gas flow rate, it is possible to calculate the pore size distribution and the mean pore size of the material.

For the characterization of the electrical properties of the developed OECT structure, a special testing setup was replicated (Figure 2) per the setup of Gualandi et al. [26]. Thereby, one voltage source controls the voltage of the source electrode whereas the other one controls the gate voltage. The multimeter measures the current output at the drain electrode of the OECT. For the setup, the voltage source PeakTech 6210 and the multimeter PeakTech 4000 were used. Two different measurements were carried out to characterize the electrical properties of the OECT. Firstly, only the source/drain voltage (V_sd_) was applied while no voltage was exerted on the gate electrode. The source/drain voltage was applied in 0.1 increments from −0.5 up to +0.5 V, and the corresponding source/drain current (I_sd_) was recorded in μA. Secondly, the gate voltage (V_g_) was applied whereas the source/drain voltage was held constant at +0.5 V. Voltage was applied to the gate electrode in steps of 0.5 V from −1.0 up to +1.0 V, and the changing source/drain current was recorded by the multimeter.

## 3. Results

The optical characterization reveals the successful impregnation of the nonwoven material with PEDOT:PSS. Also, the layered structure of the readily assembled OECT becomes visible. Moreover, the electrical measurements prove the working principle of the one-component OECT, and saline solution was detected. Lastly, also the color changing of the PEDOT:PSS-coated nonwoven textile due to the switching of oxidation states was observed. For the pore size measurement, three samples of uncoated and coated nonwoven materials were measured, and the range of the mean pore size diameter is given. The uncoated nonwoven material had a mean pore size of 18.7 to 21.9 µm, whereas the PEDOT:PSS-coated material had a slightly smaller mean pore size, between 14.5 and 17.4 µm. As expected, the material was still porous and could be used for the construction of textile-based OECTs.

### 3.1. Optical Characterization of PEDOT:PSS-Coated Nonwoven Material 

As the OECT should function not only as a transistor but also as a sensor, the uppermost electrode needs to be porous and permeable for the analytes, which can be proven by optical characterization. Figure 3 shows the comparison of the uncoated nonwoven material to the PEDOT:PSS-coated nonwoven material in different magnifications. The porous structure of the nonwoven material becomes visible at first glance as cavities between the single-fiber strands are visible in all four micrographs. Whereas the single fiber strands are simply entangled in Figure 3A, those fiber strands are glued together over large areas in Figure 2. This finding becomes even more visible in the micrographs with 100 times magnification (Figure 3A,B), where the PEDOT:PSS clearly covers and bonds the single strands. In Figure 3B, large areas covered by PEDOT:PSS are visible, while other parts remain uncoated and reveal the porous structure of the nonwoven fabric. Those open cavities may promote the diffusion of an analyte through the later source/drain electrode, while electrical conductivity is ensured over the whole surface.

Figure 4 shows the EDX analysis of the uncoated (Figure 4A,C) and PEDOT:PSS-coated (Figure 4B,D) nonwoven material. Figure 4A,B depict all elements found during the EDX analysis at 200 times magnification of the samples. The elements carbon (C: red), oxygen (O: green), and nitrogen (N: pink) can be explained by the organic origin of the nonwoven material as well as of the PEDOT:PSS. Of interest is the element sulfur (S: blue), which is a characteristic component of PEDOT:PSS, but not of the cellulosic nonwoven material. As expected, the PEDOT:PSS-coated sample (Figure 4B) has a clear blue coloring, while the blue color is nearly non-existing in the uncoated sample (Figure 4A). Figure 4C,D map only the element sulfur and prove the above-described finding; sulfur is detectable in the sample coated with PEDOT:PSS but not in the uncoated sample.

Table 1 shows the atomic percentage of the samples and supports the observation; the atomic percentage of sulfur increases from 0.68 At. -% in the uncoated sample to 4.02 At. -% in the PEDOT:PSS-coated sample. The presence of sodium in the coated sample cannot be explained, but looking at the atomic percentage of 0.66 At. -%, its presence seems to be very little and can be neglected for further consideration.

### 3.2. Optical Characterization of Readily Assembled OECT

Figure 5 shows SEM images of the readily assembled OECT with a focus on the interface, where all three components converge. The interwoven structure on the left side of the images clearly represents the ripstop woven nickel–copper fabric. The PEDOT:PSS-coated nonwoven material is visible on the right side and recognizable by the fibrous structure. In between both components, a small line of electrolyte layer becomes visible that glues together the conductive fabric and the PEDOT:PSS-coated nonwoven material. The electrolyte forms a homogenous layer showing no holes or bubbles that might lead to short circuits in the component.

Figure 6 shows the EDX mapping of the assembled OECT and the following elements are detectable: nickel (Ni: pink), copper (Cu: green), carbon (C: blue), and chlorine (Cl: turquois). Figure 6A shows the mixed mapping of all elements and a distinct separation between nickel and copper (pink–green) on the left side and carbon and chlorine (blue–turquois) on the right side becomes visible. That separation will be clarified even more when looking at the images in Figure 6B–E. Nickel (Figure 6B) and copper (Figure 6C) are primarily present on the left side of the sample, which is indicated by far more colored dots. Blue (Figure 6D) and turquoise (Figure 6E) dots are mainly present on the right side of the sample, which indicates the presence of carbon and chlorine.

These findings are as expected since nickel–copper-coated ripstop fabric has been used for the gate electrode and thus those elements should be detectable. The presence of carbon, and especially chlorine, can be explained when looking at the formulation of the electrolyte coating. Glycerol and the ECG gel, the main components of the electrolyte coating, are mainly made of carbon, oxygen, and hydrogen, which explains the presence of carbon mostly on the right side of the sample. Additionally, lithium chloride was added to the electrolyte solution and the detected chlorine in the sample can be attributed to that. Lithium is a light element and thus has a low energy of characteristic radiation so it is hardly detectable in EDX mapping. The same applies to hydrogen. Looking at the findings of Figure 4, one would expect sulfur to be present in the EDX mapping of the whole transistor, since the PEDOT:PSS-coated nonwoven material has been integrated into the OECT structure as the source/drain electrode. The non-existence of sulfur can only be explained when assuming that the PEDOT:PSS-coated nonwoven material is completely saturated and covered with the electrolyte solution. Although the PEDOT:PSS-coated nonwoven material was merely placed into the still wet electrolyte solution and not coated with it, the nonwoven material, in a sense, soaked up the solution and thereby the electrolyte solution also penetrated to the upper side of the nonwoven fabric. EDX mapping can only detect the surface of a sample but not the underlying layers, so in this case only carbon and chlorine were detectable for the electrolyte and source/drain structure.

The cross-section (Figure 7) was examined to demonstrate the layered structure of the OECT and prove the successful separation of the gate electrode and source/drain electrode by the hardened electrolyte. Thereby, it is noticeable that the electrolyte settles into the valleys and cavities of the textile electrodes so that the layer thickness of the electrolyte varies. Two measurement points are indicated that visualize the differences in layer thickness. Moreover, it must be stated, that the layer thickness of the hardened electrolyte decreased strongly compared with the wet-film thickness of 500 µm applied with the hand squeegee. This can be attributed to the loss of water during drying.

### 3.3. Electrical Characterization

Figure 8 shows the results of the electrical measurement applying source/drain voltage in steps of 0.1 V starting at −0.5 V and increasing up to +0.5 V. Simultaneously, the source/drain current (Isd) was measured as it is the characteristic output for the transistor. The measurement was carried out by recording the data for 30 s before changing the applied voltage. It can be stated that the current (I_sd_) increased from 2.26 μA at 0.0 V up to −70.66 μA when −0.5 V was applied (modulus). The same tendency can be observed for a positive voltage, though the current varied a little, whereas the current Isd amounts to −70.66 μA at −0.5 V, it was only +66.24 μA at +0.5 V. The measurement proves that the current flowed proportional to the applied voltages, which is as expected since there is an ohmic contact.

Figure 9 shows the OECT measurement with constant source/drain voltage at +0.5 V and varying gate voltage from −1.0 up to +1.0 V. Again, the source/drain current I_sd_ was measured, and data of 30 s effective testing time was recorded.

Firstly, it can be noted that the current at a gate voltage of 0.0 V complies with the current measured when V_sd_ is varied. For both measurements, the current I_sd_ at source/drain voltage of +0.5 V and no gate voltage lies at +66.24 μA in Figure 8 and +67.97 μA in Figure 9. Moreover, the influence of the applied gate voltage becomes evident. At the same source/drain voltage (V_sd_ = +0.5 V), the current output I_sd_ of the transistor can vary from +93.12 to +30.52 μA only by applying different gate voltages (V_g_ (1) = −1.0 V, V_g_ (2) = +1.0 V). It is therefore approximately reduced threefold; less current flowed between the source/drain electrode when a positive gate voltage was applied. The results show that the gate voltage can control the current that flows between the source and the drain electrode, and hence, the device operates as a transistor.

### 3.4. Color-Changing Effect of PEDOT:PSS

PEDOT:PSS is known for its color-changing properties depending on the doping state of the conductive polymer. The color palette of PEDOT:PSS ranges from dark blue to light blue to white, or colorless. PEDOT:PSS appears white to transparent in its doped state, which means that it is more conductive. In its dedoped state, PEDOT:PSS appears in a blue color, and its conductive properties are decreased [28,29,30]. As described above, applying a positive gate voltage changes the doping state of PEDOT:PSS by injecting ions in the form of cations from the electrolyte into the conductive polymer. Thereby, the negatively charged anions of PEDOT:PSS become saturated and the number of mobile holes decreases. PEDOT:PSS gets dedoped, takes on a darker color and the electrical properties deteriorate. This process is reversible and, upon applying a negative gate voltage, PEDOT:PSS gets doped again, which increases the conductivity and changes the color back to light blue or white. This effect has been observable during the electrical measurements as well and became especially visible when switching the gate voltage between −1.0 and +1.0 V.

Figure 10 shows photographs taken during the measurement and it is visible that the color of the PEDOT:PSS-coated nonwoven material slightly changed. The pictures on the left (Figure 10A,C) represent the samples when −1.0 V was applied, and thereby PEDOT:PSS was switched to the doped state. The coated nonwoven material appears brighter. The right side (Figure 10B,D), on the other hand, shows photographs of the dedoped PEDOT:PSS-coated nonwoven material, which was achieved by applying a positive gate voltage of +1.0 V. The nonwoven material took on a darker color.

### 3.5. Saline Sensing

A conceivable application of OECTs is as biosensors and many studies already discuss the use of OECTs as sensors for the detection of saline content, adrenaline, or glucose in diverse body fluids [24,26,31,32,33,34,35,36,37,38]. Therefore, the herein-developed OECTs were examined for their working behavior in presence of a 0.9% saline solution. For that, 0.09 g sodium chloride (NaCl) was mixed with 10 g distilled water. The mixing ratio is based on the salt content in the human body and hence serves as a simple indicator for use in near-body applications. A 10 μL saline solution was dripped on the source/drain surface (Figure 11), while both the gate and source/drain voltage were set at 0.5 V, respectively.

The current output (I_sd_) was simultaneously measured. Figure 12 shows the result of the measurement of one OECT over time. Figure 13, on the other hand, represents the mean value of all data of five OECTs with no saline solution added and 10 μL of 0.9% saline solution added. In Figure 12, a clear and steep drop is visible when the saline solution was added, so that the current output I_sd_ changes from approximately +41 μA to +25 μA. Less current flowed between the source and drain electrode, and the salinity can be detected by that. Also in Figure 13, the sensing mechanism of the produced OECTs is visible as the mean value of all measured OECTs is reduced from approximately 35 to 24 µA when the saline solution was added. Thereby, the standard deviation amounts to 4.7 (no solution added) and 4.76 (0.9% saline solution added) and is hence less than the actual change in current output caused by the addition of saline solution.

Next to its detection function, the ability to measure different concentrations is also of importance for a reliable sensor. Therefore, another trial was carried out whereby different concentrations of saline solution (0–2.0%) were pipetted onto the OECT and the current at the source/drain electrode was simultaneously measured (Figure 14). The source/drain and gate voltage were set at 0.5 V as before in the other experiments. To start, the source/drain current was measured when no solution was added at all and lies at 32.48 µA. Subsequently, a solution with no saline content (0.0%), 0.5%, 1.0%, and 2.0% saline content was added and drying of the device was allowed before applying the next solution. The sensor also reacted to the added solution when no saline content was present therein (19.61 µA). However, the reaction of the OECT to a solution containing saline was even stronger and increased further with rising concentration, at 0.5% saline solution current was 16.78 µA. Ultimately, the current at the source/drain electrode was reduced to 11.89 µA when 2.0% saline solution was added.

## 4. Discussion

For the production of textile OECTs, common textile technologies (impregnating and laminating) were used. Thereby, a one-component OECT was manufactured from solely textile materials. This simple manufacturing process enables the integration of the device into garments or other smart textiles, where the transistor can function as a sensor, switch, or amplifier.

The optical characterization of the PEDOT:PSS-coated nonwoven material reveals that it is possible to create a porous, yet coated, structure with this rather simple immersion process. This porosity was also proven by determining the mean pore size diameter, which was slightly reduced to a mean pore size range of 14.5 to 17.4 µm after coating. The presence of sulfur in the EDX data can be traced back to PEDOT:PSS. The optical analysis of manufactured OECTs shows a clear layering of the different components of the transistor—source/drain electrode, gate electrode, and electrolyte layer. The EDX mapping identifies characteristic elements of the single components, especially nickel and copper for the gate electrode, and also chloride that has been added as LiCl to the electrolyte.

A measurement setup has been replicated to analyze the transistor characteristics by applying different gate voltages (V_g_) and measuring the change in current (I_sd_) at the source/drain electrode. Measurements were carried out at five different OECTs and all of them functioned as a typical transistor, where the gate voltage controls the current that flows between the source and the gate electrode. Due to that, the current at the source/drain electrode varied from +93.12 μA at a gate voltage of −1.0 V to +30.52 μA at a gate voltage of +1.0 V at a constant source/drain voltage of +0.5 V. The conductivity decreased, which can be proven by Ohm’s law: *R* = *UI*. When inserting the values, the following results are derived:$$R\;\left(V_{g}\left(1\right)=-1.0\right)=\frac{0.5\;V_{sd}}{93.12\;µA}\;\approx 5.37\;k\Omega$$
$$R\;\left(V_{g}\left(1\right)=+1.0\right)=\frac{0.5\;V_{sd}}{30.52\;µA}\;\approx 16.38\;k\Omega$$

The calculations prove that a positive gate voltage (V_g_) increases the resistance of the transistor and hence reduces its conductivity. As stated in the introduction, OECTs based on PEDOT:PSS work in depletion mode, which means that in the absence of a positive gate voltage or when applying a negative gate voltage, a hole current flows in the channel, and conductivity is increased. However, when applying a positive gate voltage, positively charged cations are injected from the electrolyte into the channel and recombined with the negatively charged anions of the PEDOT:PSS. This recombination reduces the number of mobile holes in the channel and the conductive polymer becomes dedoped, the drain current drops as a result [16]. This mechanism is reinforced even more when adding an ionic substance (0.9% saline solution) as an analyte to the transistor, so that the drain current is reduced from approximately +36 to +24 μA at constant source/drain and gate voltage of +0.5 V. The resistance amounts to 13.89 kΩ when no analyte is added and increases to 18.52 kΩ under the influence of an ionic substance (0.9% saline solution). The analyte gives additional ions to the channel that interacts with the holes in PEDOT:PSS structure, and this ultimately leads to a further reduction of conductivity. Also, the quantifiable measurement of saline concentration was proven as different concentrations changed the current flow at the source/drain electrode. For a reliable sensor, specificity must be conferred on the OECT by modifying the gate electrode, electrolyte, or both. In that way, N. Coppedè et al. managed to carry out real-time measurements of both saline and adrenaline concentration in real human sweat [28]. Other literature demonstrates that OECTs are able to selectively detect glucose by adding an enzyme to the electrolyte or using different gate materials [32,33,34,35,36,37,38]. It is conceivable to adapt the OECT in such a way, that it can selectively detect bacteria [39], glucose [9], dopamine [40], lactate [41], and many more in human physiological fluids, such as blood, sweat, or saliva. Moreover, stability and durability need to be ensured, which is why additional experiments must be carried out to investigate the sensory properties of the textile OECT after twisting, bending, folding, etc. comparable to the investigations of A. Yang et al. (2018) [24].

To summarize, the measurements show that, firstly, the gate voltage controls the current output and, secondly, ionic substances can be detected by that OECT structure. Moreover, the doping and dedoping of PEDOT:PSS has not only become visible by the measured current but also by the color changing effect observed during the experiments. Figure 10 visually captures that effect. The given data proves the working principle of the herein-developed textile one-component OECTs.

## 5. Conclusions

Many textile-based organic electrochemical transistors have been reported that are based on yarn substrates that must be precisely arranged in a textile structure and are connected by a liquid electrolyte. Herein, a one-component OECT was developed that is easily integrable in textile fabrics as a solid electrolyte layer that connects both electrodes. The manufacturing of this device is based on a simple impregnation process to coat a nonwoven textile with PEDOT:PSS and a laminating process to put together the gate electrode, electrolyte, and source/drain electrode. The OECT demonstrates typical transistor characteristics, meaning that the gate voltage controls the current output. At a constant source/drain voltage of +0.5 V and a gate voltage of −1.0 V, the current is at +93.12 µA (I_sd_). When increasing the gate voltage to +1.0 V, the current drops down to +30.52 µA (I_sd_), which proves that the herein-developed transistor works in depletion mode. Moreover, saline solution was detected by decreasing the current output when the solution was applied. The easy manufacturing method, the integrability, and the results suggest that the use of these OECTs as biosensors in near-body applications is promising. However, future work must be carried out to ensure reproducibility, selectivity, durability, and full textile integration.

## Figures and Tables

**Figure 1 micromachines-13-01980-f001:**
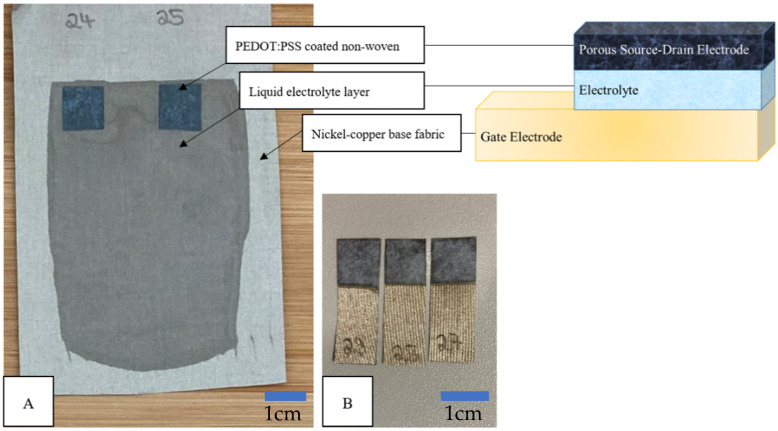
OECT Assembly. (**A**) Coating and assembling of nickel–copper base fabric with electrolyte solution and PEDOT:PSS-coated nonwoven. (**B**) Final OECT composed of the nickel–copper base fabric, hardened electrolyte layer, and PEDOT:PSS coated nonwoven fabric.

**Figure 2 micromachines-13-01980-f002:**
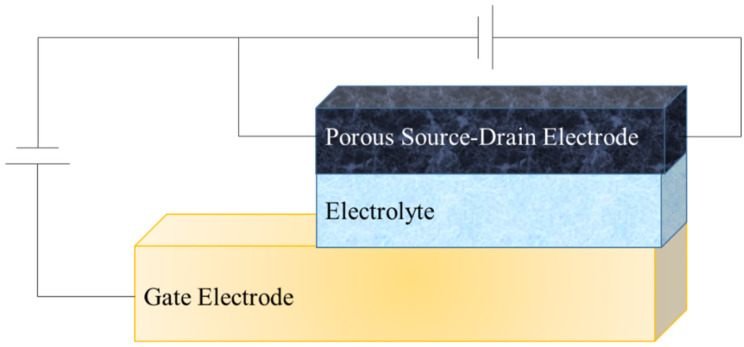
Measurement setup for electrical characterization of textile-based one-component OECT.

**Figure 3 micromachines-13-01980-f003:**
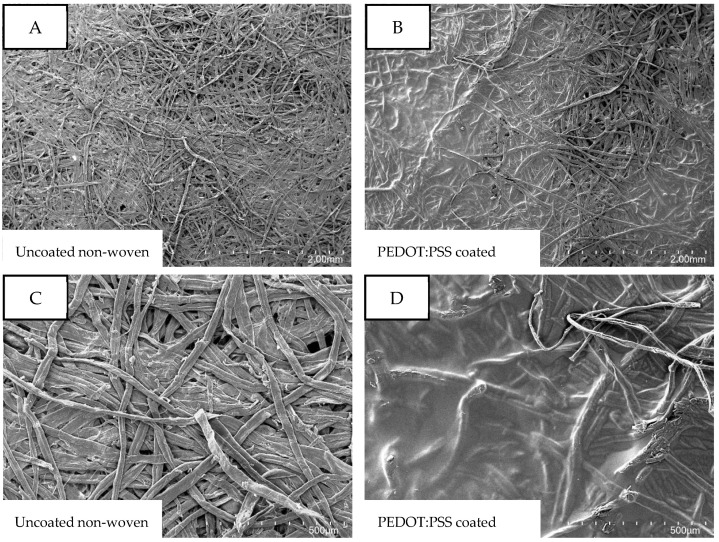
SEM images of (**A**,**C**) uncoated and (**B**,**D**) PEDOT:PSS-coated nonwoven material. (**A**,**B**): 25 times magnification. (**C**,**D**): 100 times magnification.

**Figure 4 micromachines-13-01980-f004:**
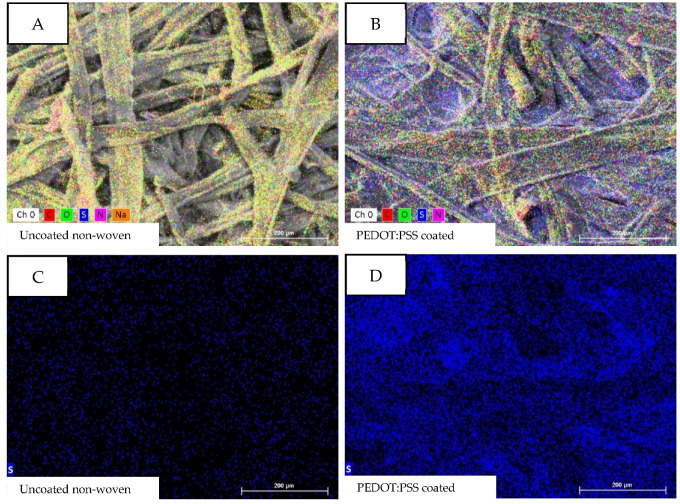
EDX analysis of (**A**) uncoated and (**B**) PEDOT:PSS-coated nonwoven material with special attention to the element sulfur in the uncoated (**C**) and coated sample (**D**).

**Figure 5 micromachines-13-01980-f005:**
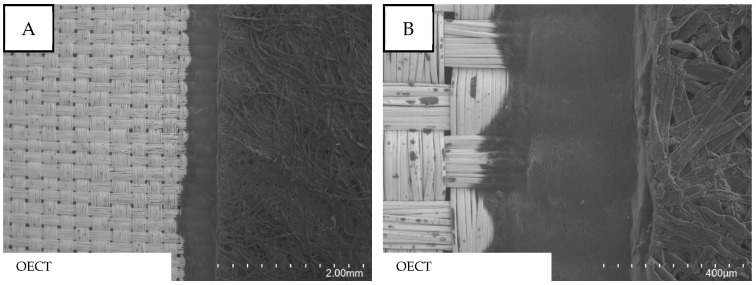
SEM images showing the layered structure of the readily assembled OECT. (**A**): 25 times magnification. (**B**): 120 times magnification.

**Figure 6 micromachines-13-01980-f006:**
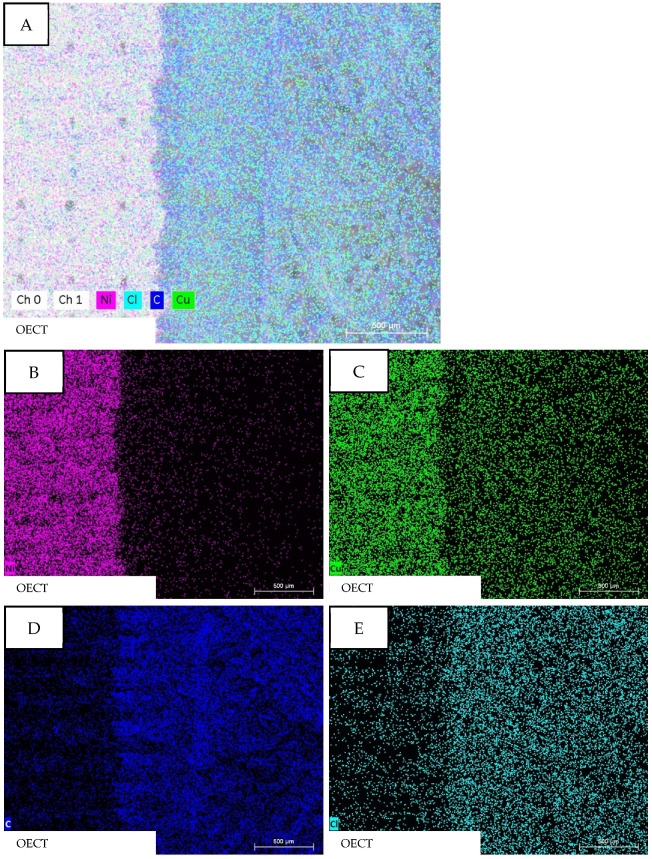
EDX mapping the elementary differences between the gate electrode and source/drain electrode of the readily assembled OECT. (**A**): All elements. (**B**): Nickel. (**C**): Copper. (**D**): Carbon. (**E**): Chloride.

**Figure 7 micromachines-13-01980-f007:**
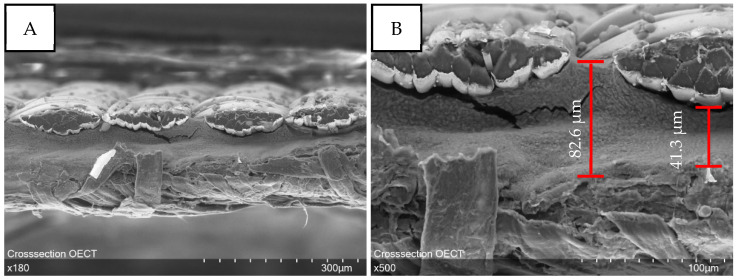
Cross-sectional examination of the layered OECT structure shows how the gate electrode and source/drain electrode are glued together by the electrolyte. (**A**): 180 times magnification. (**B**): 500 times magnification.

**Figure 8 micromachines-13-01980-f008:**
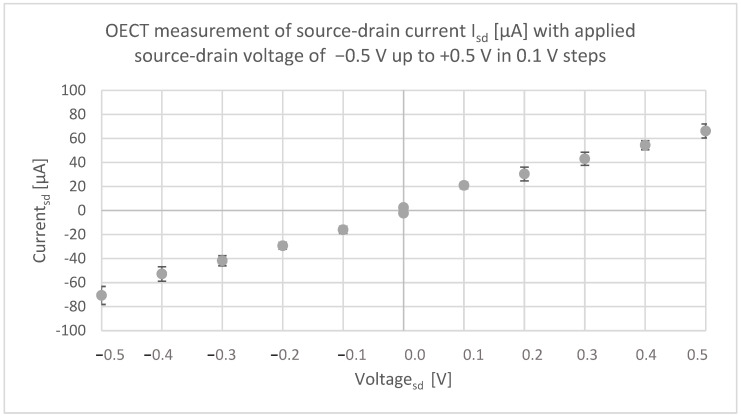
First OECT measurement of source/drain current I_sd_ [µA] showing the characteristic curve of a transistor when the V_sd_ is increased from −0.5 to +0.5 V.

**Figure 9 micromachines-13-01980-f009:**
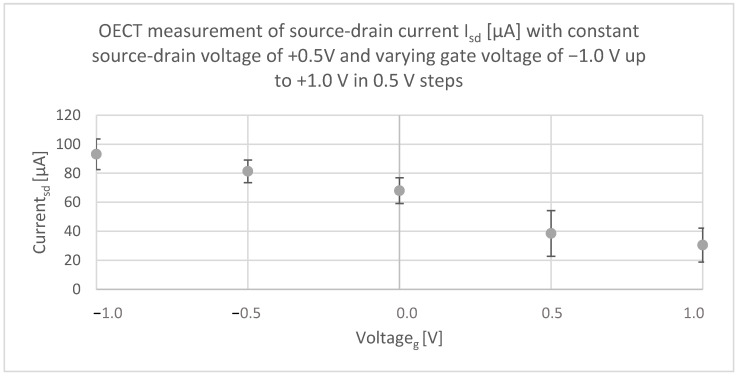
OECT measurement showing that gate voltage (V_g_) controls the current that flows between the source and drain electrode (I_sd_). When V_g_ is increased from −1.0 V to +1.0 V, the drain current drops.

**Figure 10 micromachines-13-01980-f010:**
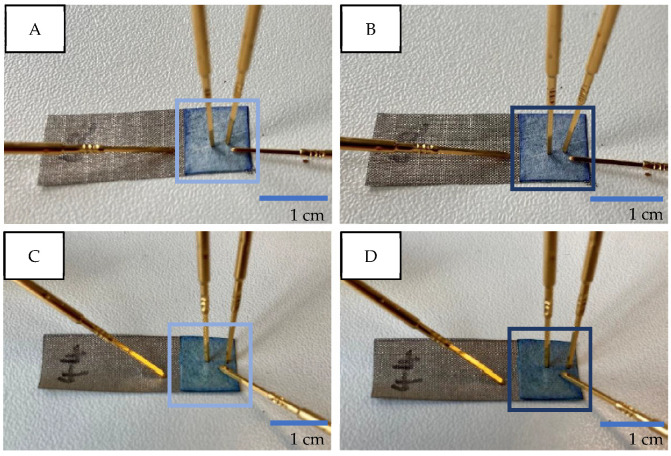
Images of color-changing effect of PEDOT:PSS upon applying different gate voltages (Vg). (**A**,**C**): −1.0 V. (**B**,**D**): +1.0 V.

**Figure 11 micromachines-13-01980-f011:**
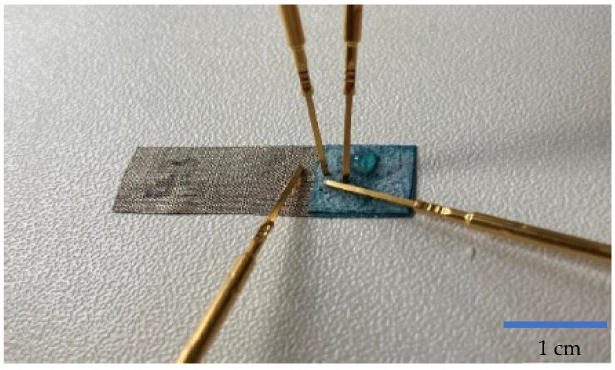
Detection of salt solution on the surface of source/drain electrode.

**Figure 12 micromachines-13-01980-f012:**
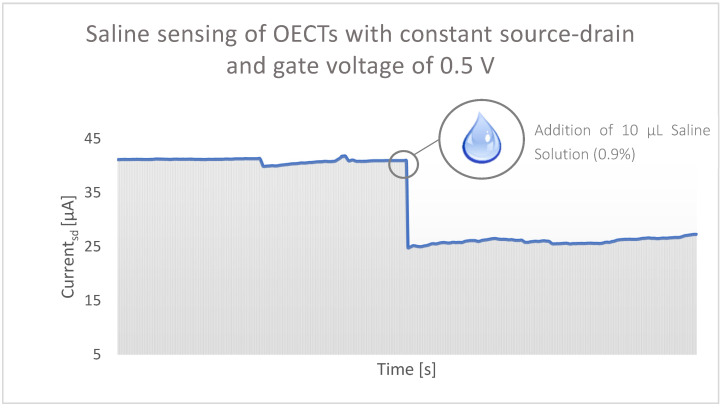
Change in current output under the addition of saline solution over time.

**Figure 13 micromachines-13-01980-f013:**
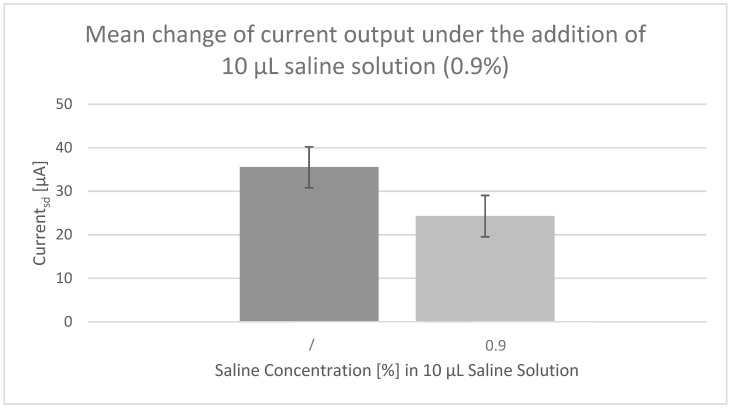
Mean change in current output under the addition of 10 µL saline solution (0.9%).

**Figure 14 micromachines-13-01980-f014:**
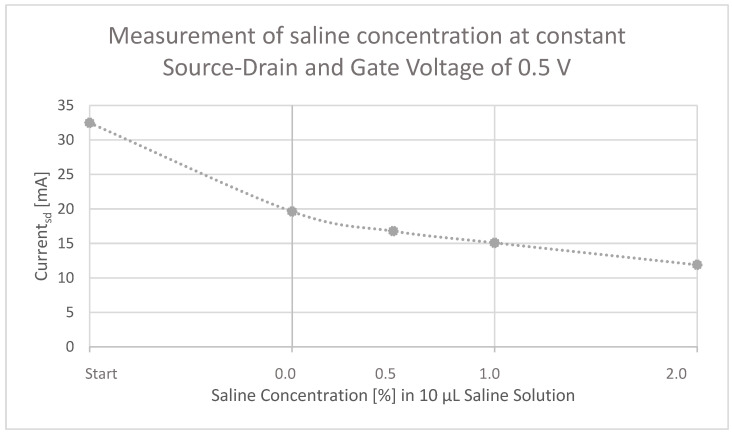
Measurement of different saline concentrations (0–2.0%) at constant source/drain and gate voltage of 0.5 V, showing that the OECT device is not only able to detect but also to measure saline concentration.

**Table 1 micromachines-13-01980-t001:** EDX-data of uncoated and PEDOT:PSS-coated nonwoven material, showing an increase of atomic mass of sulfur.

Sample	C [At. -%]	O [At. -%]	N [At. -%]	S [At. -%]	Na [At. -%]
Uncoated	51.57 ± 5.03	46.06 ± 6.01	1.69 ± 0.42	0.68 ± 0.08	/
Coated	59.33 ± 4.51	34.15 ± 3.53	1.84 ± 0.39	4.02 ± 0.27	0.66 ± 0.08

## Data Availability

Not applicable.

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
