# Peer review of "Textile One-Component Organic Electrochemical Sensor for Near-Body Applications"

_micromachines, 2022, doi:10.3390/mi13111980_

Round 1

Reviewer 1 Report

This manuscript reports a textile-based biosensor based on organic electrochemical transistor (OECT). The produced textile-based OECT exhibit all-fabric configuration, and can function as a switch, an amplifier or a sensor to detect ionic analytes. It shows the promise in utilizing textile-based OECT for wearable sensing and wearable circuit applications. Below concerns need to be addressed before the consideration on the publication in Micromachines.

1.       The author mentioned that fabric-based OECT has been developed previously (Sci. Rep. 2016, 6, 33637). What is the novelty of this work/structure compared to the published one?

2.       A scheme showing the assembled structure of the Nickel fabric and PEDOT:PSS-nonwoven is suggested. Scale bars of all the photographs should be included.

3.       To evaluate the permeability of the electrode, it is better to test the air/moisture permeability value, rather than the microscopic observation.

4.       The PEDOT:PSS-coated nonwoven was laminated onto the Nickel fabric by the electrolyte. How was the adhesion stability? Will the source electrode be pill off when it is subjecting to ionic analyte?

5.       How was the durability of the devices? Is it possible to integrate such all-textile-based devices into garments?  

6.       The citation is a little bit out of date. 

Author Response

Dear Reviewer,

Thank you for your input and thoughtful comments. We would like to answer all of them point by point.

  1. The author mentioned that fabric-based OECT has been developed previously (Sci. Rep. 2016, 6, 33637). What is the novelty of this work/structure compared to the published one?

-> Most organic electrochemical transistors on textile materials are yarn-based. Thereby, one yarn is coated with PEDOT:PSS to function as a source-drain electrode with channel. Another electro active yarn functions as a gate electrode and those yarns need to be assembled in a distinct structure; clearly separated and only connect by the electrolyte, which was liquid in all found literature. In here, we developed a layered structure, so that you do not have two separate yarns but one textile component that is connected by a hardened electrolyte.

We tried to make it more clear in the abstract and introduction so we hope the readers will understand more easily the novelty of this device layout.

  1. A scheme showing the assembled structure of the Nickel fabric and PEDOT:PSS-nonwoven is suggested. Scale bars of all the photographs should be included.

-> We added a schematic drawing to show the correspondence between the different layers and the components of a transistor in Figure 1. Also, we added the missing scale bars on the photographs.

  1. To evaluate the permeability of the electrode, it is better to test the air/moisture permeability value, rather than the microscopic observation.

-> Thank you for that input. We will consider it for further investigations but just now all porous PEDOT:PSS coated non-wovens are installed in the OECT component, so testing the air/moisture permeability is impossible.

  1. The PEDOT:PSS-coated nonwoven was laminated onto the Nickel fabric by the electrolyte. How was the adhesion stability? Will the source electrode be pill off when it is subjecting to ionic analyte?

-> As we did not test the adhesion, we cannot make any statements about the stability. The source-drain electrode did not pill off when it was subjected to the ionic analyte. Because of the commercial application of the non-woven (filtering paper), it was considered to be stable when coming in contact with liquid medium.

  1. How was the durability of the devices? Is it possible to integrate such all-textile-based devices into garments?  

-> Unfortunately, we did not test the durability of the devices so far, so we cannot make any statements about that.

As we used common textile technologies and materials that are easily processable by those technologies, it will be possible to integrate those OECTs into garments. We added that point in the discussion. 

  1. The citation is a little bit out of date. 

-> We added some more literature, but unfortunately especially the literature dealing with textile based OECTs is all we could find.

Reviewer 2 Report

This paper demonstrated the capacity in detecting saline solution of proposed textile-based transistor. However, as an electrochemical sensor, this device is far away from satisfaction. Several issues should be addressed.

1. The characterization of the relationship between ion concentration and output current is not referred. The shown results only proved its detection function. What about the ability of measuring the concentration parameters, which is more important for a sensor?

2. About the selectivity. This problem has been mentioned as a future work in the conclusion, but it is not enough for a sensor. According to the working principle, different electrolytes, maybe, can trigger a similar response in the sensor. How to focus the measuring ability on a certain ion? How to distinguish the target ion? Does this device feature the potential of simultaneously detecting different targets?

There are also some comments on the paper writing.

1. In Subsection 2.1.2, filtering paper and non-woven are simultaneously mentioned. Do they refer to a same material?

2. The OECT are made by assembling different layers. It is better to add a schematic diagram to show the correspondence between the layers and structure of transistor.

3. Similar work is also needed in the experiment section to show how to apply the voltage to the OECT. Four probes are used, then schematic diagrams are needed to show their functions.

4. A sectional view of the OECT prototype is needed to show the thickness of each layer and their combining state.

5. The caption of Fig. 10 should be improved to show more details of each subfigure. The state of OECT before voltage should be given in this figure.

6. The introduction should be further improved to show more recent works on using transistor as electrochemical sensors.

7. I cannot well understand the world “One-Component” in paper title. Many flexible transistors are made on only a piece of substrate. What the difference between yours and other transistors when considering the world “One-Component”?

Author Response

Dear Reviewer,

Thank you for your time reading our manuscipt and thoughtful feedback. We would like to answer your comments point by point.

  1. In Subsection 2.1.2, filtering paper and non-woven are simultaneously mentioned. Do they refer to a same material?

-> Yes, it refers to the same material and we put it in brackets to make it clearer to the reader.

  1. The OECT are made by assembling different layers. It is better to add a schematic diagram to show the correspondence between the layers and structure of transistor.

-> We added a schematic drawing to show the correspondence between the different layers and the components of a transistor in Figure 1.

  1. Similar work is also needed in the experiment section to show how to apply the voltage to the OECT. Four probes are used, then schematic diagrams are needed to show their functions.

-> We added a schematic drawing on how the voltage is applied/current measured. Also in section 2.2.4 we mention literature on that our measurement set-up is based (Gualandi et al.)

  1. A sectional view of the OECT prototype is needed to show the thickness of each layer and their combining state.

 -> We added a sectional view of the layered OECT structure in Figure 6.

  1. The caption of Fig. 10 should be improved to show more details of each subfigure. The state of OECT before voltage should be given in this figure.

-> We added some details to the captions. The state before voltage is applied is given at 0.0 V in Figure 8 and 9.

  1. The introduction should be further improved to show more recent works on using transistor as electrochemical sensors.

-> We added some more literature on OECTs in bio sensing applications.

  1. I cannot well understand the world “One-Component” in paper title. Many flexible transistors are made on only a piece of substrate. What the difference between yours and other transistors when considering the world “One-Component”?

-> As it also says in the title, the herein presented OECT is based on textile materials, which is the most important part of our work. Most organic electrochemical transistors on textile materials are yarn-based.  Thereby, one yarn is coated with PEDOT:PSS to function as a source-drain electrode with channel. Another electro active yarn functions as a gate electrode and those yarns need to be assembled in a distinct structure; clearly separated and only connect by the electrolyte, which was liquid in all found literature. In here, we developed a layered structure, so that you do not have two separate yarns but one textile component that is connected by a hardened electrolyte.

We tried to make it more clear in the abstract and introduction so we hope the readers will understand more easily the novelty of this device layout.

Best regards
Rike Brendgen

Reviewer 3 Report

Brendgen et al. reported a textile-based organic electrochemical sensor for near body applications. After carefully reviewing this manuscript, I think it is not suitable for publication in Micromachines.

1.     Authors need to carefully revise this manuscript from pictures, logic, date, application and so on to make it a scientific paper.

2.     For example, Figure 4 should be Figure 2. This very obvious error indicates that the author did not carefully proofread before submitting.

Author Response

Dear Reviewer,

Thank you for taking the time reading our manuscript. We hope that after the changes, you like it better.

  1. Authors need to carefully revise this manuscript from pictures, logic, date, application and so on to make it a scientific paper.

-> We are sorry to hear that you disliked our manuscript. We hope though, that after the changes you like the manuscript better.

  1. For example, Figure 4 should be Figure 2. This very obvious error indicates that the author did not carefully proofread before submitting.

-> We are really sorry for that unpleasant mistake! Of course, we changed the caption of the Figures to the right order.

Best regards

Rike Brendgen

Round 2

Reviewer 1 Report

The reviewer still has the following concerns.

1. There is fabric-based OECT in the literature. Sci. Rep. 2016, 6, 33637 reported a fabric-based OECT instead of a yarn-based OECT. The author didn't answer the novelty compared to this OECT. To the reviewer, in-plane device with all electrodes embedded in the fabric seems to have better stability. 

2. The reviewer doesn't think measuring air and moisture permeability is impossible. As far as it is a layered fabric structure, moisture permeability can be measured.

3. The stability of the device can be reflected by if the device could function well after bending/twisting/folding. The reviewer thinks this is an important property talking about E-textile applications.

Author Response

Dear Reviewer,

First of all, we would like to apologize for the length of time it took to edit your corrections. Unfortunately, me and my colleague got sick, so we weren't able to work at all for a few weeks. But lately, we were able to carry out the missing experiments and found time to work on your reviews.

Let me answer you point by point:

  1. There is fabric-based OECT in the literature. Sci. Rep. 2016, 6, 33637 reported a fabric-based OECT instead of a yarn-based OECT. The author didn't answer the novelty compared to this OECT. To the reviewer, in-plane device with all electrodes embedded in the fabric seems to have better stability. 

-> In the introduction we tried to clarify the novelty of the device layout. Compared to the one-mentioned (Sci. Rep. 2016, 6, 33637), we use a hardened electrolyte that is anchored in the device structure, so that our textile based OECT can be used as an amplifier/switch without additionally applied electrolytes.

  1. The reviewer doesn't think measuring air and moisture permeability is impossible. As far as it is a layered fabric structure, moisture permeability can be measured.

-> We determined the pore size of the non-woven by measuring air permeability and added that part in section 2.2.4.

  1. The stability of the device can be reflected by if the device could function well after bending/twisting/folding. The reviewer thinks this is an important property talking about E-textile applications.

-> We discussed your point in section 4 and will definitely make comparable experiments in our future work with OECTs to test longevity, stability and durability of those flexible textile based devices.

Thank you for your time and insightful contributions.

Reviewer 2 Report

The authors only referred parts of my comments, but ignore the more important ones. Therefore, major revision is still needed to address the following comments.

1. The characterization of the relationship between ion concentration and output current is not referred. The shown results only proved its detection function. What about the ability of measuring the concentration parameters, which is more important for a sensor?

2. About the selectivity. This problem has been mentioned as a future work in the conclusion, but it is not enough for a sensor. According to the working principle, different electrolytes, maybe, can trigger a similar response in the sensor. How to focus the measuring ability on a certain ion? How to distinguish the target ion? Does this device feature the potential of simultaneously detecting different targets?

3. About the novelty. Many transistors on fabric have been reported (e.g. the one given by other reviewer), it is necessary to show the real contribution of your paper.

Author Response

Dear Reviewer,

First of all, we would like to apologize for the length of time it took to edit your corrections. Unfortunately, me and my colleague got sick, so we weren't able to work at all for a few weeks. But lately, we were able to carry out the missing experiments and found time to work on your reviews.

Let me answer you point by point: 

  1. The characterization of the relationship between ion concentration and output current is not referred. The shown results only proved its detection function. What about the ability of measuring the concentration parameters, which is more important for a sensor?

-> We carried out another measurement with different saline concentrations. The device is able to measure different concentrations. The result is shown in Figure 14.

  1. About the selectivity. This problem has been mentioned as a future work in the conclusion, but it is not enough for a sensor. According to the working principle, different electrolytes, maybe, can trigger a similar response in the sensor. How to focus the measuring ability on a certain ion? How to distinguish the target ion? Does this device feature the potential of simultaneously detecting different targets?

-> We discussed that point in section 4. The electrolyte, gate electrode or both need to be adjusted to the analyte to be detected. Selectivity will be part of our new study about textile OECTs but unfortunately we did not analyze it just yet.  

  1. About the novelty. Many transistors on fabric have been reported (e.g. the one given by other reviewer), it is necessary to show the real contribution of your paper.

-> In the introduction we tried to clarify the novelty of the device layout. Compared to the one-mentioned (Sci. Rep. 2016, 6, 33637), we use a hardened electrolyte that is anchored in the device structure, so that our textile based OECT can be used as an amplifier/switch without additionally applied electrolytes.

Thank you for your time and insightful contributions.

Best regards

Rike Brendgen

Reviewer 3 Report

The authors made changes to the manuscript, and I agreed to publish it in Micromachines.

Author Response

Dear Reviewer,

First of all, we would like to apologize for the length of time it took to edit your corrections. Unfortunately, me and my colleague got sick, so we weren't able to work at all for a few weeks. But lately, we were able to carry out the missing experiments and found time to work on your reviews.

Thanks for agreeing to publish the manuscript in micromachines, your time and insightful contributions.

Best regards

Rike Brendgen

Round 3

Reviewer 2 Report

The revison work is good, and the paper can be accepeted.

Meantime, I am looking forward to your next work on the device with great selectivity.